# MORE: MULTI-FRAME OUTDOOR POINT CLOUD REGISTRATION WITH TEMPORAL MEMORY BUFFER

## ABSTRACT

Existing outdoor point cloud registration methods are commonly constrained by the pairwise input paradigm, which neglects sufficient temporal information intrinsically within consecutive LiDAR sequences. In this paper, we propose a novel **M**ulti-frame **O**utdoor point cloud **R**egistration network with t**E**mporal memory buffers (**MORE**). The key observation is that long-term temporal LiDAR sequences can provide rich global contextual information to complete sparse measurements, filter outliers, and address low-overlap problems, which further boosts registration performance. Specifically, two memory buffers are designed, including both the implicit memory feature buffer and explicit memory pose buffer, to store and dynamically update temporal pose-related features. We further leverage a Mamba-based temporal encoding layer to effectively integrate current motion features with history motion features. Moreover, a novel dynamic history weighting module is developed to adaptively rescale current and history pose-related features. Extensive experiments on three outdoor datasets demonstrate the superiority of MORE, surpassing all previous state-of-the-art methods by 32% RTE and 17% RRE reduction on KITTI, 37% RTE and 4% RRE reduction on nuScenes, and 29% RTE and 9% RRE reduction on Apollo-Southbay. Our method also generalizes well to the multiview indoor point cloud registration task with rather competitive performance on 3DMatch, 3DLoMatch, and ScanNet datasets. Codes will be released upon publication.

## 1 INTRODUCTION

Point cloud registration (Besl & McKay, 1992) is a fundamental task in the computer vision field, which aims to find the optimal transformation matrix between point cloud frames. It is widely applied in various downstream tasks, such as autonomous driving (Lu et al., 2021; Xue et al., 2024) and SLAM systems (Liu et al., 2023a; Deng et al., 2024).

Even though object-level and indoor point cloud registration have been widely explored (Huang et al., 2021; Qin et al., 2022; Yew & Lee, 2022; Ao et al., 2023), few works have investigated the large-scale outdoor registration task (Lu et al., 2021; Liu et al., 2023b). Challenges are mainly three-fold: (1) Distinct from object-level point cloud, outdoor points are typically more sparse, irregular, noisy, and have a wider spatial distribution range (Lu et al., 2023b). (2) Outliers, e.g., occlusions and highly dynamic objects, introduce inconsistent motion patterns, undermining the final ego-motion regression (Liu et al., 2023b). (3) Low-overlap point pairs pose another significant challenge (Huang et al., 2021). Due to the high speed of ego vehicles in autonomous driving, consecutive LiDAR scans often have large translational displacements, making it difficult to search for neighboring points across frames (Xue et al., 2024). HRegNet (Lu et al., 2021) addresses mismatches and ensures reliable correspondences caused by outliers with bilateral and neighborhood consensus in point-matching layers. RegFormer (Liu et al., 2023b) uses a projection-aware Transformer with enlarged receptive fields to handle the low-overlap issue. *However, these prior outdoor registration methods only use pairwise inputs, ignoring the rich temporal information in consecutive LiDAR sequences.*

Inspired by recent advances in using temporal information for tasks like semantic segmentation (Li et al., 2023; Lao et al., 2023; Chen et al., 2023), object detection (Wang et al., 2023b; Hou et al., 2024), and multi-object tracking (Gao & Wang, 2023), we aim to improve registration accuracy by extending pairwise frames to multi-frame inputs. The key motivations are: (1) Long-term temporal

information helps complete scenes by supplementing sparse and noisy observations across consecutive frames. (2) Multi-frame inputs enhance temporal consistency, improving the recognition and rejection of outliers for effective point correlation. (3) For the low-overlap problem, motion priors from past timestamps can provide strong pose initialization and reduce the distance between initially distant point pairs. These advantages will be proved in the experiment section.

In this paper, we propose a novel multi-frame point cloud registration network, MORE, designing temporal memory buffers and dynamic history weighting to explore inherent temporal consistency as in Fig. 1. Temporal information from multi-frame history timestamps is progressively retrieved and stored to improve each pairwise registration performance, rather than directly outputting multi-frame poses. Two memory buffers are designed to store implicit temporal features (*Memory Feature Buffer*) and explicit temporal transformations (*Memory Pose Buffer*) with a fixed length. Specifically, when the current pair of frames is delivered into the network, the Memory Feature Buffer is first updated by concatenating the current motion features and prestored multi-frame history motion features. Then, a Mamba-based (Gu & Dao, 2023) temporal en-

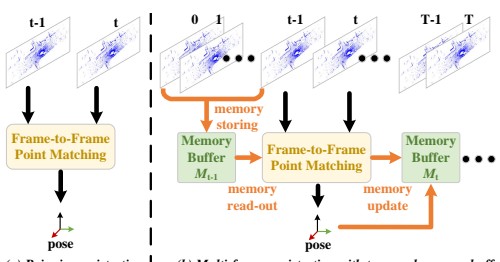

Figure 1: **Comparison with previous pairwise registration methods.** Previous pairwise outdoor registration methods often lack sufficient temporal exploration in LiDAR sequences. In contrast, we propose a multi-frame registration network that leverages long-term temporal information through memory buffers.

coding layer is developed to establish long-term temporal dependencies across the updated memory features with high efficiency. Similarly, temporally stored poses from previous timestamps are processed and correlated by another Mamba module in the Memory Pose Buffer, generating transformation features. To adaptively merge history and current pose-related features, a dynamic history weighting module inspired by (Jia et al., 2016; Aydemir et al., 2023) is designed to rescale history and current features and adaptively weight current with historical multi-frame features. Finally, the current pose is regressed and refined, which is used to update memory buffers for the next timestamp.

Overall, the key contributions of this paper are:

- We propose a novel outdoor point cloud registration paradigm enhanced by multi-frame temporal motion information, which improves the robustness to challenges like outliers and low-overlap inputs.

- Explicit and implicit memory buffers store temporal information from successive frames. Memory Feature Buffer holds implicit pose-related features, while Memory Pose Buffer maintains explicit history poses for enhanced pose initialization. To integrate history and current features, a dynamic history weighting module is designed to adaptively merge features on different scales.

- Our method achieves state-of-the-art performance on outdoor registration datasets, including KITTI (Geiger et al., 2012), nuScenes (Caesar et al., 2020), and Apollo-Southbay (Lu et al., 2019). Furthermore, our method generalizes well to the indoor multiview registration task on 3DMatch, 3DLoMatch (Zeng et al., 2017), and ScanNet Dai et al. (2017).

## 2 RELATED WORK

**Outdoor Point Cloud Registration.** Point cloud registration (Zeng et al., 2017; Deng et al., 2018; Choy et al., 2019; Aoki et al., 2019; Dong et al., 2020; Qin et al., 2022; Mu et al., 2024; Jia et al., 2024; Chen et al., 2024b; Yuan et al., 2024; Jiang et al., 2024; Fu et al., 2025) is a crucial task that targets at finding the optimal transformation matrix between point cloud frames. Recently, large-scale outdoor registration has raised remarkable research attention, but the original sparsity, irregularity, and high-dynamics characteristics of outdoor LiDAR points pose significant challenges. As a pioneering work, HRegNet (Lu et al., 2021) proposes bilateral consensus and neighborhood consensus to reduce mismatches. RegFormer (Liu et al., 2023b) designs a projection-aware Transformer, incorporating the cross-attention and all-to-all point-gathering strategy for reliable cross-frame correlation. To address density disparities, GCLNet (Liu et al., 2023c) introduces a group-wise con-

trastive learning to extract density-invariant geometric features for registration. SDMNet (Lu et al., 2023a) develops a sparse-to-dense registration network with a sparse matching stage and a local-dense matching stage. Darls (Wang et al., 2025a) designs a density adaptive registration method for large-scale point clouds. However, these previous works typically take pairwise-only point pairs and neglect the inherent temporal consistency within LiDAR sequences. In this paper, we propose a novel outdoor registration model enhanced by multi-frame temporal consistency.

**Temporal Modeling with Memory.** Sequential 4D point cloud streams have great potential for boosting various dense prediction tasks (Wang & Tian, 2024). Different from a static point cloud with limited observations, the additional temporal dimension can supplement the incomplete measurements (Li et al., 2023), display consistent motion priors (Gao & Wang, 2023), and further boost the performance of original single-frame tasks (Wang & Tian, 2024). Recently, increasing research has focused on storing temporal information with a memory mechanism. STMM (Xiao & Lee, 2018) develops a recurrent unit for memory storage for object detection, modeling the long-term temporal appearance and motion dynamics. MeMOT (Cai et al., 2022) designs a memory-based multi-object tracking framework that utilizes a long-term spatio-temporal memory storing the identity features of tracked objects. MemorySeg (Li et al., 2023) proposes a 3D latent memory representation to improve the current predictions with temporal information from past frames. Inspired by these works, we design both explicit and implicit memory buffers for temporal information storage.

## 3 METHOD

### 3.1 OVERALL ARCHITECTURE

Given a stream of LiDAR point cloud sequences $\{PC_0, PC_1, ..., PC_{V-1}\} \in \mathbb{R}^{V \times N \times 3}$, we propose a multi-frame registration framework as described in Fig. 2, where buffers storing multiple history memories would enhance the performance of each pairwise registration. Specifically, to correlate current point pairs $PC_t$ and $PC_{t+1}$, hierarchical point features are first extracted, and then matched in the coarsest layer following HRegNet (Lu et al., 2023b). Afterwards, history motion information is read-out from memory buffers on both feature level and pose level in Sec. 3.2. To dynamically merge history and current features, we also design a dynamic history weighting module in Sec. 3.3. Finally, the estimated poses are iteratively refined, generating the final rotation matrix $\hat{R} \in \mathbb{R}^{3 \times 3}$ and translation vector $\hat{t} \in \mathbb{R}^{3 \times 1}$. Memory buffers are also updated with current features and current pose in Sec. 3.4. The network is supervised by the loss function as in Sec. 3.5.

### 3.2 MEMORY INITIALIZATION AND READ-OUT

Unlike pairwise inputs in previous registration works, point cloud videos are the inputs of our network for long-term temporal modeling. To take advantage of temporal motion priors sufficiently, we design two memory buffers storing both cross-frame implicit feature-level memory and explicit pose-level memory. Our memory buffers $M_0^F \in \mathbb{R}^{T \times D_f}$ of features and $M_0^P \in \mathbb{R}^{T \times D_P}$ of poses are initialized by empty sets, where the temporal length of the memory buffers is maintained as $T$.

**Memory Read-out.** With motion features from the current timestamp $F_t$ and poses $R_t$, $t_t$ from the coarse matching layer in HRegNet (Lu et al., 2023b), we first read-out the temporally-stored features using a sliding window mechanism. When current features are delivered into the Memory Feature Buffer $M_{t-1}^F$, the temporally-farthest feature in the buffer $\hat{F}_{t-T-1}$ is discarded. Other $T-1$ temporally-nearest features $\{\hat{F}_{t-T}, ..., \hat{F}_{t-1}\}$ are concatenated with the current motion feature $F_t$ as:

$$M_{t-1}^F : \{\hat{F}_{t\text{-}T\text{-}1}, \hat{F}_{t\text{-}T+1}, ..., \hat{F}_{t\text{-}1}\} \Rightarrow \{\hat{F}_{t\text{-}T}, ..., \hat{F}_{t\text{-}1}, F_t\}, \tag{1}$$

where $\{\}$ indicates the feature concatenation along the temporal dimension. The concatenated features formulate the input tokens of the following Mamba-based Temporal Encoding:

$$\hat{M}_{t-1}^F = \text{LN}(M_{t-1}^F), \tag{2}$$

$$\overline{M}_t^F = \delta(\text{DW}(\text{Linear}(\hat{M}_{t-1}^F))), \tag{3}$$

$$\hat{M}_t^F = \delta(\text{Linear}(\hat{M}_{t-1}^F)), \tag{4}$$

$$\hat{F}_t = \text{Linear}(\text{SSM}(\overline{M}_t^F)) \odot \hat{M}_t^F + M_{t-1}^F, \tag{5}$$

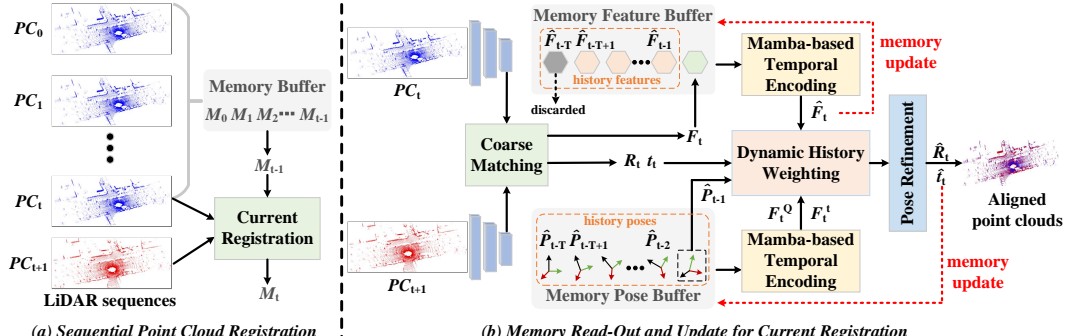

Figure 2: **The overview architecture of MORE.** We leverage the long-term temporal features from history frames to enhance the current pairwise registration performance, as in (a). During the current registration, memory features and memory poses are first read-out and interacted with current motion features through the Mamba-based Temporal Encoding module in (b). Then, the current refined poses will in turn update contents in the memory buffers.

where $\delta$ denotes the SiLU activation (Hendrycks & Gimpel, 2016). DW is the depth-wise convolution (Chollet, 2017). LN indicates the Layer Normalization. SSM indicates the standard selective state space model (Gu & Dao, 2023). The output feature $\hat{F}_t$ is the temporally encoded feature.

For the Memory Pose Buffer, rotation matrices $M_{t-1}^{R} = \{\hat{Q}_{t-T-1}, \hat{Q}_{t-T}, ..., \hat{Q}_{t-1}\}$ and translation matrices $M_{t-1}^{t} = \{\hat{t}_{t-T-1}, \hat{t}_{t-T}, ..., \hat{t}_{t-1}\}$ from the $T$-length nearest memory pose buffer $M_{t-1}^{P}$ are respectively embedded by two separate MLP layers first for lifting their dimensions. To facilitate regression, the rotation matrix here is represented by quaternion vectors $\hat{Q}_t \in \mathbb{R}^{4 \times 1}$. Finally, the embedded rotation features and translation features are fed into two Mamba blocks for the temporal modeling, respectively:

$$F_t^Q = \text{Mamba}(\text{MLP}(\{\hat{Q}_{t-T-1}, \hat{Q}_{t-T}, ..., \hat{Q}_{t-1}\})), \tag{6}$$

$$F_t^t = \text{Mamba}(\text{MLP}(\{\hat{t}_{t-T-1}, \hat{t}_{t-T}, ..., \hat{t}_{t-1}\})), \tag{7}$$

where $\hat{Q}_t$ and $\hat{t}_t$ respectively denote the $t$-th temporally-nearest quaternion and translation vectors estimated from history frames in the Memory Pose Buffer. Here, Mamba indicates the same operation as in Memory Feature Buffer, for which we avoid detailed descriptions for simplicity.

## 3.3 DYNAMIC HISTORY WEIGHTING

After retrieving the history-interacted features, the key problem is how to merge the history-interacted features with the current motion features for enhanced registration performance. Because there may be scale gaps, the history and current poses should be re-scaled for effective information merging. Inspired by the great success of the dynamic weighting mechanism in (Jia et al., 2016; Aydemir et al., 2023), a dynamic history weighting layer is introduced to adaptively merge the history pose-related features, history poses, current pose-related features, and current poses.

To be specific, the inputs of our dynamic history weighting layer are composed of the history pose $\hat{Q}_{t-1}, \hat{t}_{t-1}$ from the last previous timestamp, history pose-encoded features $F_t^Q, F_t^t$, current pose $R_t, t_t$, and current pose-encoded features $\hat{F}_t$. The current rotation matrix $R_t$ is also converted to the quaternion vector $Q_t$ for consistency. Here, we only give descriptions of the quaternion vector weighting, and the same goes for the translation matrix weighting. History and current pose-related features are first concatenated as:

$$F_d^Q = \hat{Q}_{t-1} \oplus F_t^Q \oplus Q_t \oplus \hat{F}_t, \tag{8}$$

where $\oplus$ indicates concatenation along the channel dimension. Then, the dynamic weighting head is designed as:

$$W_1 = W_{d_1} F_d^Q, \; W_2 = W_{d_2} F_d^Q, \tag{9}$$

$$F_{d_1}^Q = \text{ReLU}(\text{Norm}(W_1 F_d^Q)), \; Q_t^d = W_2 F_{d_1}^Q, \tag{10}$$

where $W_{d_1}$ and $W_{d_2}$ are trainable parameters, and the output dimension is set to the same as the quaternion vector. Similar processes are conducted to generate the weighted translation vector $t_t^d$. By providing the history pose-related features to the dynamic prediction head as inputs, the dynamic weights enable the network to adjust to different scales dynamically and re-weight current pose estimations with history motion priors.

## 3.4 MEMORY UPDATE AND POSE REFINEMENT

Our fixed-length memory buffers follow the sliding window mechanism, which is progressively updated as consecutive point frames are delivered one by one into the network. For the Memory Feature Buffer update, we directly concatenate the encoded current features $\hat{F}_t$ with $T$-1 previous features:

$$M_t^{\mathrm{F}} = \{\hat{F}_{t\text{-}T+1}, ..., \hat{F}_{t\text{-}1}, \hat{F}_t\}. \tag{11}$$

For the Memory Pose Buffer update, first, the pose refinement in (Lu et al., 2023b) is adopted to refine the poses $Q_t^d$ and $t_t^d$ to get the final pose estimation $\hat{R}_t$ and $\hat{t}_t$. The Memory Pose Buffer is then updated by concatenating previous $T$-1 poses and the current pose estimations:

$$M_t^{\mathrm{R}} = \{\hat{R}_{t\text{-}T+1}, ..., \hat{R}_{t\text{-}1}, \hat{R}_t\}, \tag{12}$$

$$M_t^{\mathrm{t}} = \{\hat{t}_{t\text{-}T+1}, ..., \hat{t}_{t\text{-}1}, \hat{t}_t\}. \tag{13}$$

## 3.5 LOSS FUNCTION

Our network is supervised by reducing the discrepancy for each pairwise registration samples (Lu et al., 2023b; Liu et al., 2023b). The rotation and translation loss functions are:

$$\mathcal{L}_{\mathrm{rot}} = ||\hat{R}^T R - \mathrm{I}||_2, \mathcal{L}_{\mathrm{trans}} = ||\hat{t} - t||_2, \tag{14}$$

where $\hat{R}, \hat{t}$ indicate the estimated rotation and translation matrices. $R, t$ are the ground truth rotation and translation matrices. I denotes the identity matrix. The overall loss function is: $\mathcal{L} = \mathcal{L}_{\mathrm{trans}} + \mathcal{L}_{\mathrm{rot}}$.

# 4 EXPERIMENT

## 4.1 EXPERIMENTAL SETTINGS

**Datasets.** We conduct the experiments on three commonly used outdoor LiDAR datasets: KITTI odometry (Geiger et al., 2012), nuScenes (Caesar et al., 2020), and Apollo-Southbay (Lu et al., 2019). KITTI consists of 11 sequences with ground truth pose annotations. Following the settings in (Lu et al., 2021; Liu et al., 2023b), we use sequences 00-05 for training, 06-07 for validation, and 08-10 for evaluation. To guarantee a fair comparison, we follow (Lu et al., 2021; Xue et al., 2024) to refine the noisy ground truth labels using the ICP algorithm (Besl & McKay, 1992) and construct the input point pairs with an interval of 10 frames, creating more challenging registration inputs. nuScenes is composed of 1,000 scenes: 850 scenes are leveraged for training and validation, and the other 150 scenes are for evaluation. Apollo-Southbay dataset provides official training/ testing splits. We follow these splits and use only valid point clouds for registration.

**Evaluation Metrics.** Following the protocols in previous works (Lu et al., 2023b;a; Liu et al., 2023b), we use three main metrics for the evaluation of our method: Relative Translation Error (RTE), Relative Rotation Error (RRE), and Registration Recall. A successful registration is decided only when the RTE and RRE are both within certain thresholds $\theta_T, \theta_R$. To avoid the unreliable error metric report by minor failure samples, we follow previous works (Lu et al., 2021; Xue et al., 2024; Liu et al., 2023b) to calculate the average RTE and RRE on successful registration samples.

**Implementation Details.** We first voxelize the input points with a voxel size of 0.3 m. Following (Lu et al., 2021; 2023a; Xue et al., 2024), down-sampling is adopted for the input points, where 16,384 points are randomly sampled for KITTI and Apollo-Southbay, and 8,192 points for nuScenes. Experiments are conducted on a single NVIDIA RTX3090 GPU. The Adam optimizer is adopted with $\beta_1 = 0.9$, $\beta_2 = 0.999$. The initial learning rate is 0.001 and exponentially decays every 200,000 steps until reaching 0.00001. The batch size is set to 8. We choose HRegNet* (Lu et al., 2023b)

Table 1: Outdoor point cloud registration performance on the KITTI dataset (Geiger et al., 2012). '*' indicates the baseline on which we introduce multi-frame temporal information.

| Methods | KITTI Odometry | | | |
|---|---|---|---|---|
| | RTE (m) | RRE (deg) | Recall | Time (ms) |
| ICP (P2P) (Besl & McKay, 1992) | 0.045 ± 0.054 | 0.112 ± 0.093 | 14.25% | 472.2 |
| ICP (P2Pl) (Besl & McKay, 1992) | 0.044 ± 0.041 | 0.145 ± 0.153 | 33.56% | 461.7 |
| RANSAC (Fischler & Bolles, 1981) | 0.126 ± 0.067 | 0.538 ± 0.396 | 91.94% | 459.4 |
| FGR (Zhou et al., 2016) | 0.929 ± 0.592 | 0.963 ± 0.806 | 39.42% | 506.1 |
| IDAM (Li et al., 2020) | 0.659 ± 0.483 | 1.057 ± 0.939 | 70.92% | 33.4 |
| FMR (Huang et al., 2020) | 0.657 ± 0.422 | 1.493 ± 0.847 | 90.57% | 85.5 |
| DGR (Choy et al., 2020) | 0.322 ± 0.319 | 0.374 ± 0.302 | 98.71% | 1496.6 |
| HRegNet (Lu et al., 2021) | 0.056 ± 0.075 | 0.178 ± 0.196 | 99.77% | 106.2 |
| RegFormer (Liu et al., 2023b) | 0.082 ± 0.112 | 0.228 ± 0.209 | 99.80% | 98.3 |
| OKR-Net (Wang et al., 2023c) | 0.070 ± 0.130 | 0.230 ± 0.220 | 99.22% | 100.4 |
| HRegNet* (Lu et al., 2023b) | 0.047 ± 0.037 | 0.147 ± 0.120 | **100.0%** | 136.0 |
| HDMNet (Xue et al., 2024) | 0.050 ± 0.057 | 0.159 ± 0.152 | 99.85% | 120.2 |
| PTT (Wang et al., 2025b) | 0.063 | 0.230 | 99.80% | - |
| LSReg-Net (Tao et al., 2025) | 0.060 ± 0.070 | 0.200 ± 0.170 | 99.92% | 92.5 |
| Ours | **0.032 ± 0.036** | **0.122 ± 0.096** | **100.0%** | 133.0 |

Table 2: Outdoor point cloud registration performance on the nuScenes dataset (Caesar et al., 2020). '*' indicates the baseline on which we introduce multi-frame temporal information.

| Methods | nuScenes | | | |
|---|---|---|---|---|
| | RTE (m) | RRE (deg) | Recall | Time (ms) |
| ICP (P2P) (Besl & McKay, 1992) | 0.252 ± 0.510 | 0.253 ± 0.502 | 18.78% | 82.0 |
| ICP (P2Pl) (Besl & McKay, 1992) | 0.153 ± 0.296 | 0.212 ± 0.306 | 36.8% | 44.5 |
| RANSAC (Fischler & Bolles, 1981) | 0.206 ± 0.186 | 0.738 ± 0.704 | 60.91% | 187.4 |
| FGR (Zhou et al., 2016) | 0.708 ± 0.622 | 1.007 ± 0.924 | 32.2% | 284.6 |
| DCP (Wang & Solomon, 2019) | 1.087 ± 0.491 | 2.065 ± 1.141 | 58.58% | 45.5 |
| IDAM (Li et al., 2020) | 0.467 ± 0.410 | 0.793 ± 0.783 | 87.98% | 32.6 |
| FMR (Huang et al., 2020) | 0.603 ± 0.391 | 1.610 ± 0.974 | 92.06% | 61.1 |
| DGR (Choy et al., 2020) | 0.211 ± 0.183 | 0.476 ± 0.430 | 98.41% | 523.0 |
| HRegNet (Lu et al., 2021) | 0.122 ± 0.112 | 0.273 ± **0.197** | **100.0%** | 87.3 |
| RegFormer (Liu et al., 2023b) | 0.198 | 0.223 | 99.90% | 85.6 |
| OKR-Net (Wang et al., 2023c) | 0.140 ± 0.130 | 0.280 ± 0.210 | 99.96% | 60.0 |
| HRegNet* (Lu et al., 2023b) | 0.110 ± 0.096 | 0.285 ± 0.209 | **100.0%** | 120.1 |
| HDMNet (Xue et al., 2024) | 0.114 ± 0.102 | 0.274 ± 0.206 | **100.0%** | 102.9 |
| LSReg-Net (Tao et al., 2025) | 0.150 ± 0.130 | 0.320 ± 0.220 | 99.98% | 81.2 |
| Ours | **0.069 ± 0.055** | **0.261 ± 0.197** | **100.0%** | 119.3 |

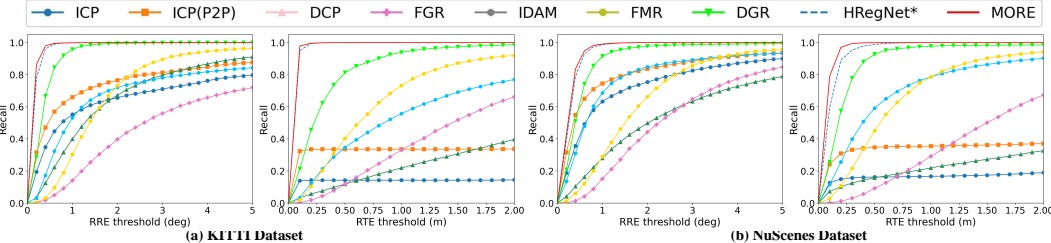

Figure 3: **The registration recall under different RRE or RTE thresholds on the KITTI and nuScenes datasets.**

here as the baseline, where '*' indicates the baseline method on which we introduce multi-frame temporal information. Registration is successful when $\theta_T = 5$m, $\theta_R = 2$ degree (deg) by default. The temporal length in buffers $T$ is set to 20 for KITTI and Apollo-Southbay, and 10 for nuScenes.

## 4.2 QUANTITATIVE RESULTS

To demonstrate the effectiveness of our proposed method, we compare with both optimization-based methods (Besl & McKay, 1992; Zhou et al., 2016; Fischler & Bolles, 1981) and learning-based methods (Wang & Solomon, 2019; Li et al., 2020; Huang et al., 2020; Choy et al., 2020; Lu et al., 2021; 2023b; Xue et al., 2024; Wang et al., 2023c) on three datasets.

**Comparison Results on KITTI.** As in Table 1, our model not only outperforms the previous optimization methods, e.g., ICP (Besl & McKay, 1992), RANSAC (Fischler & Bolles, 1981), by a large margin but also the learning-based methods. Compared to the previous SOTA method HRegNet* (Lu et al., 2023b), our model achieves a 31.9% lower average RTE and a 17.0% lower average RRE enhanced by multi-frame temporal information. Also, compared to the baseline (Lu et al., 2023b), our method requires a slightly shorter runtime for one inference. The experiment results demonstrate both the accuracy and efficiency of our proposed method.

Table 3: Outdoor point cloud registration performance on the Apollo-Southbay dataset (Lu et al., 2019). '*' indicates the baseline method on which we introduce multi-frame temporal information.

| Methods | Apollo-Southbay | | | |
|---|---|---|---|---|
| | RTE (m) | RRE (deg) | Recall | Time (ms) |
| ICP (P2P) (Besl & McKay, 1992) | 0.100 ± 0.039 | 0.063 ± 0.305 | 39.93% | 482.1 |
| ICP (P2Pl) (Besl & McKay, 1992) | 0.039 ± 0.179 | 0.046 ± 0.257 | 46.45% | 470.2 |
| RANSAC (Fischler & Bolles, 1981) | 0.125 ± 0.114 | 0.361 ± 0.368 | 83.72% | 552.1 |
| FGR (Zhou et al., 2016) | 0.470 ± 0.498 | 0.664 ± 0.762 | 50.50% | 496.0 |
| DCP (Wang & Solomon, 2019) | 1.174 ± 0.499 | 2.155 ± 1.254 | 28.49% | 41 |
| IDAM (Li et al., 2020) | 0.456 ± 0.416 | 0.361 ± 0.429 | 74.03% | 32.0 |
| FMR (Huang et al., 2020) | 0.653 ± 0.488 | 0.727 ± 0.658 | 83.78% | 83.4 |
| DGR (Choy et al., 2020) | 0.132 ± 0.151 | 0.127 ± 0.146 | 99.64% | 2507.0 |
| HRegNet* (Lu et al., 2023b) | 0.034 ± 0.037 | 0.079 ± 0.079 | 99.88% | 129.5 |
| Ours | **0.024 ± 0.031** | **0.072 ± 0.074** | **100.0%** | 127.6 |

**Comparison Results on nuScenes.** As in Table 2, our method achieves a 100% registration recall and outperforms previous optimization-based and learning-based methods with smaller registration

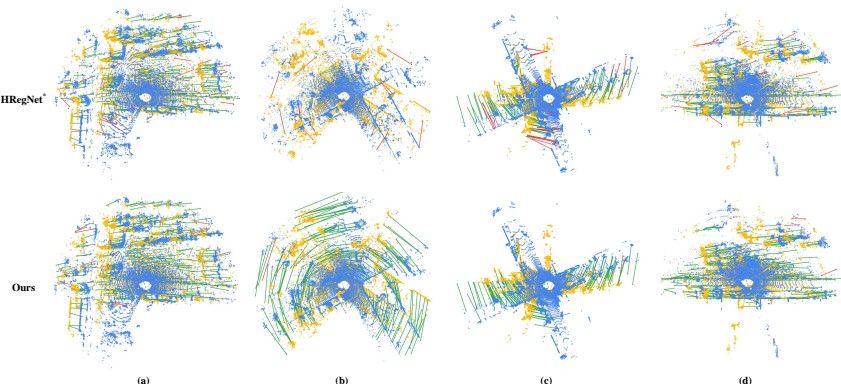

Figure 4: **The visualization of point cloud registration on KITTI.** Yellow and blue points indicate input point pair frames, with green lines showing accurate correspondences (inliers) and red lines showing inaccurate ones (outliers). This figure is best viewed in zoomed-in color mode. Compared to HRegNet* (Lu et al., 2023b), our method MORE estimates more accurate correspondences.

errors, pushing the limit of performance further. Compared to the recent SOTA method HRegNet* (Lu et al., 2023b), our model results in only half of its translation errors (0.069 m) and also a much lower rotation error (0.261 deg).

**Comparison on Apollo-Southbay.** As shown in Table 3, our method achieves 100% registration recall and significantly outperforms all previous methods, especially in the RTE metric, establishing a new SOTA in performance.

**Results Under Stricter Thresholds.** Since the threshold setting in (Lu et al., 2023b) is rather loose, registration recalls are relatively saturated in these three datasets (achieving 100%). In this case, the performance comparison with previous methods is unobvious. Therefore, we formulate a series of stricter thresholds as in Fig. 3 for KITTI and nuScenes datasets, where our method

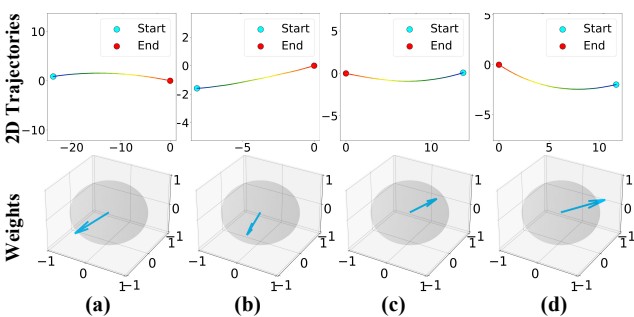

Figure 5: **Dynamic history weighting visualization.** We observe that similar history trajectories (upper figures) commonly share similar weights (lower figures), such as (a)&(b), (c)&(d). The $W_2$ matrix is projected to a normalized 3D vector via PCA.

consistently outperforms previous optimization-based and learning-based methods.

### 4.3 QUALITATIVE RESULTS

**Registration Samples.** As shown in Fig. 4, compared to HRegNet* (Lu et al., 2023b), introducing temporal information with memory buffers can significantly improve the ratio of successfully correlated keypoints. This demonstrates the effectiveness of our designed temporal information encoding for filtering outliers. More samples are visualized in the Appendix.E.

**Weights in Dynamic History Weighting.** We display the weights and corresponding 2D trajectories in Fig. 16. Similar weights are observed for similar temporal trajectories, demonstrating that our Dynamic History Weighting module can effectively learn weights from historical trajectories.

### 4.4 GENERALIZE TO MULTIVIEW INDOOR REGISTRATION

To demonstrate the generalization ability of our method, we also evaluate our method MORE on indoor-level registration datasets: 3DMatch, 3DLoMatch (Zeng et al., 2017), and ScanNet (Dai et al., 2017) in Table 4 for multiview registration. Our method outperforms recent state-of-the-art methods (Arrigoni et al., 2016; Chatterjee & Govindu, 2017; Yew & Lee, 2021; Lee & Civera, 2022;

Table 4: Generalization to multiview indoor registration on 3D(Lo)Match and ScanNet. We follow SGHR (Wang et al., 2023a) with pruned pose graph, using YOHO (Wang et al., 2022) for pairwise registration. 'Rot.' and 'Trans.' indicate rotation and translation errors.

| Method | 3DMatch Recall(%)↑ | 3DLoMatch Recall(%)↑ | ScanNet Rot. Mean/Med (°)↓ | ScanNet Trans. Mean/Med (m)↓ |
|---|---|---|---|---|
| EIGSE3 (Arrigoni et al., 2016) | 40.1 | 26.5 | 40.6/37.1 | 0.88/0.84 |
| L1-IRLS (Chatterjee & Govindu, 2017) | 68.6 | 49.0 | 41.8/34.0 | 1.05/1.01 |
| RotAvg (Chatterjee & Govindu, 2017) | 77.2 | 63.0 | 38.5/31.6 | 0.96/0.83 |
| LITS (Yew & Lee, 2021) | 80.8 | 65.2 | 24.9/19.9 | 0.65/0.56 |
| HARA (Lee & Civera, 2022) | 83.8 | 79.1 | 34.7/31.3 | 0.86/0.17 |
| SGHR (Wang et al., 2023a) | 96.2 | 81.6 | 21.7/19.0 | 0.56/0.49 |
| SMVR (Fang et al., 2024) | 96.2 | 82.0 | 19.8/17.5 | 0.55/0.45 |
| Ours | **97.1** | **82.8** | **19.2/16.3** | **0.52/0.44** |

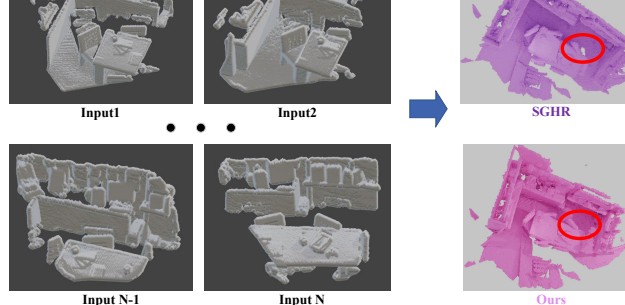

Figure 6: **Multiview indoor registration on 3DMatch (Zeng et al., 2017).** We adapt our model to indoor scenes with better performance than the baseline SGHR (Wang et al., 2023a). SGHR has inaccurate pose estimation for areas with the red circle, while ours is more accurate.

Fang et al., 2024), which are specially designed for the multiview indoor registration task. As in Fig. 6, the network inputs are a series of partial scans with overlap. Our method registers input scans better compared to SGHR (Wang et al., 2023a), demonstrating our method's strong generalization ability for diverse indoor-level scenes.

## 4.5 ABLATION STUDIES

In this section, we conduct comprehensive ablation studies to demonstrate the effectiveness of each proposed novelty.

**The Significance of Temporal Memory Buffers.** Table 5 shows ablation studies for two memory buffers. Without the Memory Pose Buffer (MPB), the average RTE increases by 31.3% and the average RRE by 27.9%. Removing the Memory Feature Buffer (MFB) results in a 9.4% higher average RTE and a 3.3% higher average RRE.

**Different Temporal Lengths.** Table 6 shows the comparison of different settings on temporal lengths for memory buffers. Accuracy increases largely from $T=5$ to 20 but is saturated when $T>20$. This is because 20 frames have enough historical observations for memory read-out. Points farther than 20 frames share less geometric and motion similarity, which may even decrease accuracy.

**Dynamic History Weighting (DHW).** As in Table 14, removing DHW results in an 18.8% higher RTE and an 11.5% higher RRE, indicating that the Dynamic History Weighting can effectively unify the history and current pose-related features. We also compare with another pose estimator in (Liu et al., 2024c), which has much higher RTE and RRE. More ablation studies about various temporal interaction methods, rotation and translation thresholds, and more visualizations are provided in the supplementary materials.

## 5 DISCUSSIONS

The temporal information introduced by our designed memory buffers can address challenging cases in which previous pairwise registration works struggle to effectively establish accurate point corre-

Table 5: Ablation studies of the two memory buffers on KITTI.

| MFB | MPB | RTE (m) | RRE (deg) | Recall | Time (ms) |
|-----|-----|---------|-----------|--------|-----------|
|  |  | $0.047 \pm 0.046$ | $0.164 \pm 0.135$ | **100%** | **126** |
| ✓ |  | $0.042 \pm 0.042$ | $0.156 \pm 0.124$ | **100%** | 130 |
|  | ✓ | $0.035 \pm 0.039$ | $0.126 \pm 0.106$ | **100%** | 132 |
| ✓ | ✓ | **0.032 ± 0.036** | **0.122 ± 0.096** | **100%** | 133 |

Table 6: Ablation studies of different temporal lengths on KITTI.

| Window lengths | RTE (m) | RRE (deg) | Recall | Time (ms) |
|----------------|---------|-----------|--------|-----------|
| T=5 | $0.037 \pm 0.041$ | $0.136 \pm 0.116$ | **100%** | **128** |
| T=10 | $0.033 \pm 0.036$ | $0.127 \pm 0.101$ | **100%** | 129 |
| T=20 | **0.032 ± 0.036** | **0.122 ± 0.096** | **100%** | 133 |
| T=30 | **0.032 ± 0.038** | $0.129 \pm 0.105$ | **100%** | 140 |

Table 7: Ablation studies of the Dynamic History Weighting (DHW) on KITTI.

| Model | RTE (m) | RRE (deg) | Recall | Time (ms) |
|-------|---------|-----------|--------|-----------|
| w/o DHW | $0.038 \pm 0.038$ | $0.136 \pm 0.113$ | **100%** | 138 |
| with estimator in (Liu et al., 2024c) | $0.036 \pm 0.040$ | $0.132 \pm 0.104$ | **100%** | 135 |
| with DHW | **0.032 ± 0.036** | **0.122 ± 0.096** | **100%** | **133** |

Table 8: Evaluation on low-overlap input pairs with 20-frame and 30-frame intervals on KITTI.

| Method | Intervals | RTE (m) | RRE (deg) | Recall |
|--------|-----------|---------|-----------|--------|
| HRegNet* (Lu et al., 2023b) | 20-frame | $0.085 \pm 0.152$ | $0.235 \pm 0.213$ | 96.2% |
| Ours | 20-frame | **0.041 ± 0.039** | **0.148 ± 0.127** | **100.0%** |
| HRegNet* (Lu et al., 2023b) | 30-frame | $0.123 \pm 0.195$ | $0.315 \pm 0.302$ | 76.3% |
| Ours | 30-frame | **0.046 ± 0.042** | **0.169 ± 0.141** | **100.0%** |

spondences. Here, we analyze three main challenging cases for the outdoor point cloud registration task: low-overlap, dynamics, and occlusions.

**Low-Overlap Inputs.** Low-overlap registration is still an open issue (Yin et al., 2024) because real-world LiDAR scans may have large distances or rotations even for consecutive frames, especially for low-frequency scanning LiDAR sensors, fast-moving cars, and street corners with large rotations. In these cases, the classical nearest neighboring strategy, e.g., KNN, may fail to capture precise point correspondences. In Table 8, we evaluate the low-overlap registration performance with larger frame intervals, which demonstrates the effectiveness of our designed memory buffers compared to the pairwise baseline method HRegNet* (Lu et al., 2023b).

**Dynamics and Occlusions.** Dynamic or occluded objects introducing inconsistent motion patterns are commonly viewed as outliers in the outdoor registration task (Liu et al., 2023b), which undermines the accuracy of consistent pose regression. We visualize the estimated correspondences for scenes with significant occlusions in Fig. 7 and high dynamics in Fig. 8. Our method can mitigate their influences with fewer or even no estimated correspondences on these outlier points. Therefore, the subsequent pose regression would be less influenced by outliers.

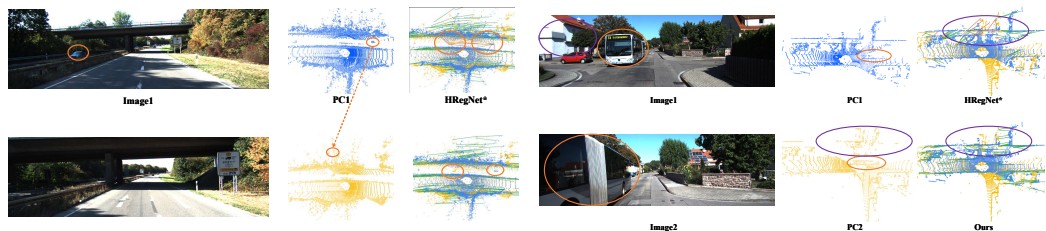

Figure 7: **Comparison on dynamics.** We evaluate on highway scenes, including high dynamic cars from the opposite direction (orange circled). HRegNet* (Lu et al., 2023b) wrongly estimates correspondences on dynamics, while ours can filter influence with no correspondence estimation on dynamics.

Figure 8: **Comparison on scenes with significant occlusions.** For scenes with large missing areas occluded by the bus (orange circled). HRegNet* (Lu et al., 2023b) wrongly estimates correspondences for occluded objects (purple circled), but ours can filter their influence with no correspondences on occlusion.

## 6 CONCLUSION

In this paper, we introduce a memory mechanism to enhance pairwise registration performance using multi-frame history features for outdoor point cloud registration. Two temporal buffers are designed to store feature-level and pose-level memories from previous timestamps, exploiting long-term history motion priors to address challenging outdoor scenes. Additionally, a Mamba-based temporal interaction module and a dynamic history weighting module are developed to merge historical and current features. Extensive experiments on the KITTI, nuScenes, and Apollo-Southbay datasets demonstrate the effectiveness of our proposed method in various driving environments.

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

APPENDIX

# A  OVERVIEW

The supplementary materials are structured as follows:

- We first provide more details about the network architecture in Sec. B.
- The evaluation metrics are described in Sec. C.
- Additional experiment results are presented in the Sec. D to demonstrate the superiority of our method and the effectiveness of the proposed contributions.
- More visualizations on the KITTI, nuScenes, and Apollo-Southbay datasets are provided in Sec. E.
- Section F discloses the limited and strictly assistive usage of a large language model (LLM) during manuscript polishing.
- Additionally, we append the video demos of the outdoor registration performance on four sequences of KITTI (Seq.01, Seq.08, Seq.09, Seq.10) to the supplementary materials with the file name `MORE_01.mp4`, `MORE_08.mp4`, `MORE_09.mp4`, `MORE_10.mp4`.

# B  NETWORK ARCHITECTURE

## B.1  FEATURE EXTRACTION AND COARSE MATCHING

We follow HRegNet (Lu et al., 2023b) to construct the hierarchical point feature extraction and coarse point matching layers.

**Feature Extraction.** Hierarchical point features are extracted as in (Lu et al., 2023b). For each layer, Weighted Farthest Point Sampling (WFPS) (Qi et al., 2017) is first utilized to sample a set of center keypoints. Around each keypoint, K Nearest Neighbor (KNN) and Shared Multi-layer Perceptron (Shared MLP) are then used to search and aggregate the features of its k nearest neighboring points:

$$f_i = \max_{k=1,2,\cdots,\mathrm{K}} \mathrm{MLP}((x_i^k - x_i) \oplus f_i^k), \tag{15}$$

where $x_i^k$ denotes the $k$-th nearest point and $f_i^k$ is the corresponding point feature. Max is the Max-pooling operation.

**Coarse Point Matching.** The core issue in point cloud registration is to find precise and reliable correspondences between two frames. At the coarsest layer, where points commonly possess the largest receptive field, finding accurate correspondences can guarantee the generally reliable registration performance. To achieve this, we consider geometric features, descriptor features, and similarity features from current point pairs as in HRegNet (Lu et al., 2023b). Specifically, for each keypoint in $PC_t$ as the clustering center, we perform KNN to select a set of neighboring candidate points in $PC_{t+1}$. The center keypoint and its K neighboring points form a cluster for feature embedding. Geometric features consist of both the absolute coordinates and the relative distance of the keypoint and its neighboring points. Descriptor features include the saliency uncertainties of keypoints and also the local descriptors for each keypoint and its neighboring points. Similarity features are considered in both bilateral consensus and neighborhood consensus. Bilateral consensus ensures the bi-directional optimal matching scores in both the forward (keypoints in $PC_t$ searching KNN in $PC_{t+1}$) and the reverse (keypoints in $PC_{t+1}$ searching KNN in $PC_t$) processes. Neighborhood consensus indicates that the neighboring keypoints around the correspondence keypoints should possess similar features. The outputs of the coarse matching layer are cross-frame features $F_t$ and initially estimated pose $R_t$, $t_t$. However, these features and poses contain no temporal information, which will interact with and update temporal memory buffers (Secs. 3.3-3.5 in the main paper).

## B.2  MAMBA ARCHITECTURE

Recently, a novel architecture named Mamba (Gu & Dao, 2023) has shown promising effectiveness, especially for long-sequence modeling ability (Li et al., 2025), outperforming Transformer-based

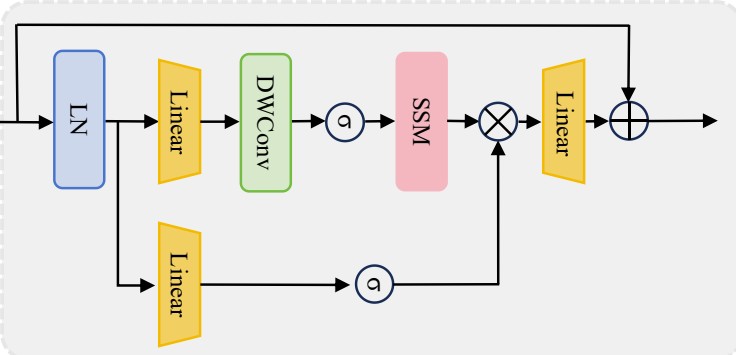

Figure 9: **The architecture of the Mamba module.** We utilize the standard Mamba (Gu & Dao, 2023) module without any task-specific design.

Table 9: **Implementation details of coarse matching and pose refinement layers.** In the coarse matching and pose refinement layers, 'N' denotes the number of the selected neighboring points in KNN. 'MLP1' and 'MLP2' represent the output dimension of two MLP layers, respectively, for correspondence prediction.

| Module | Layer | N | MLP1 | MLP2 |
|---|---|---|---|---|
| Coarse Matching | Layer 1 | 8 | [128, 128, 128] | [128, 128, 1] |

| Module | Layer | N | MLP1 | MLP2 |
|---|---|---|---|---|
| Pose Refinement | Layer 2 | 8 | [256, 256, 256] | [256, 256, 1] |
|  | Layer 3 | 8 | [512, 512, 512] | [512, 512, 1] |

counterparts in image processing (Zhu et al., 2024; Liu et al., 2024d), video understanding (Li et al., 2025; Liu et al., 2024a), and point cloud (Liang et al., 2024; Liu et al., 2024b) fields. In this paper, we adopt the vanilla Mamba architecture (Gu & Dao, 2023) in our temporal encoding layer as illustrated in Fig. 9. Without any task-specific designs, our Mamba-based temporal encoding turns out to be effective enough to embed long-term dependencies for history and current motion features.

### B.3 POSE REFINEMENT LAYER

We adopt the same pose refinement process as in HRegNet (Lu et al., 2023b), where finer registration layers only consider the geometric features and descriptor features. Similarity features are abandoned due to high computational costs. With the correlated features, a soft correspondence layer is introduced to predict keypoint correspondences and their confidence scores (Lu et al., 2021). Finally, the Weighted Kabsch algorithm is applied to calculate the transformation matrices $\Delta R$ and $\Delta t$. Without loss of generality, taking layer $l$ as an example, the refinement layer can be represented by:

$$R^l = \Delta R \times R^{l+1}, \tag{16}$$
$$t^l = \Delta R \times t^{l+1} + \Delta t, \tag{17}$$

where $R^l, t^l$ indicate the refined poses at a finer layer. $R^{l+1}, t^{l+1}$ indicate the un-refined poses at a coarser layer. More implementation details are provided in Table 9.

### C EVALUATION METRICS

Here, we provide the equations to calculate RTE (Relative Translation Error) and RRE (Relative Rotation Error). RTE is calculated by:

$$RTE = ||\hat{t} - t||_2. \tag{18}$$

RRE is defined as:

$$RRE = \arccos(\mathrm{Tr}(\hat{R}^{\mathrm{T}}R - 1)/2), \tag{19}$$

Table 10: **More comparison results on the KITTI odometry (Geiger et al., 2012).**

| Methods | KITTI Odometry | | |
| --- | --- | --- | --- |
| | RTE (m) | RRE (deg) | Recall |
| PosDiffNet (She et al., 2024) | 0.066 | 0.240 | 99.80% |
| TrT-Net (Chen et al., 2024a) | 0.063 | 0.230 | 99.80% |
| ML-SemReg (Yan et al., 2024) | 0.052 | 0.200 | 97.91% |
| Ours | **0.032** | **0.122** | **100.0**% |

where $\hat{R}, \hat{t}$ indicate the respective estimated rotation matrix and translation vector, and $R, t$ are the ground truth rotation matrix and translation vector.

## D  ADDITIONAL EXPERIMENTAL RESULTS

**More Comparison Results.** We also supplement more comparisons with recent state-of-the-art methods including PosDiffNet (She et al., 2024), TrT-Net (Chen et al., 2024a), and ML-SemReg (Yan et al., 2025). Since they only release the average rotation and translation metrics, we also follow them for a fair comparison. As shown in Table 10, our method has substantial performance advantages compared to these methods, which demonstrates the superiority of our proposed temporal buffers.

**Performance Under More Challenging Thresholds.** We also compare with baseline HRegNet* (Lu et al., 2023b) in terms of different settings $\theta_T$ and $\theta_R$ in Table 11 and Table 12. From these tables, much obviously higher recalls, lower RRE, and lower RTE can be observed compared to HRegNet*. This demonstrates the great potential of our MORE in more challenging outdoor scenes with larger rotations and translations.

**Ablation Studies for Various Temporal Interaction Methods.** For effective memory read-out, we explore different temporal interaction methods, including GRU-based, Attention-based, and Mamba-based approaches. As shown in Table 13, the Mamba-based method outperforms the others, particularly in reducing rotation errors. This improvement is due to the effectiveness of the Mamba model in capturing long-term temporal dependencies.

**Ablation Studies for Different Mamba Layers.** We also vary the number of Mamba layers from 1 to 3 in Table 14. Performance remains stable across all settings, demonstrating our model is insensitive to the specific design choice in Mamba. Notably, we choose 1 as the layer number of the Mamba module because increasing the number cannot bring a significant performance increase.

**Ablation Studies for Learnable Parameters in DHW.** Learnable weights are of great significance to the dynamic fusion of history and current pose-related features in the Dynamic History Weighting (DHW). To prove this, we replace the learnable weighting matrices $W_1, W_2$ with fixed ones. As shown in Table 15, using fixed $W_1$ or $W_2$ increases both RTE and RRE compared to the learnable DHW, showing that dynamic weighting truly works.

## E  VISUALIZATION RESULTS

**Outliers: Dynamic and Occlusion Cases.** Dynamics and occlusions are two main challenging outlier cases for outdoor point cloud registration (Liu et al., 2023b). Here, in Fig. 10, we first show one case of dynamics. As illustrated in the figure, when a dynamic car runs toward the ego-car at a high speed, parts of the dynamic car will be unobserved. For example, given only Frame 2 and Frame 3, the registration is difficult as the head of the car is not scanned in Frame 3. However, introducing more temporal inputs (Frame 1) will provide more geometric cues of the dynamic car and help recognize this outlier. This case can often be seen in the highway scenario shown in `MORE_01.mp4` appended to the supplementary materials. Another common outlier is occlusion. For example, the occluded car in Frame 2 of Fig. 11 is often seen as an outlier with pairwise inputs (e.g., Frames

Table 11: **Comparison results with HRegNet*** (**Lu et al., 2023b**) **under different translation thresholds** $\theta_T$ **on the KITTI dataset.** $\theta_R$ is set to 1deg.

| $\theta_T$ (m) | Method | RTE (m) | RRE (deg) | Recall |
|---|---|---|---|---|
| 0.05 | HRegNet* | $0.0302 \pm 0.0110$ | $0.1583 \pm 0.1155$ | 72.08% |
| | Ours | $\mathbf{0.0243 \pm 0.0109}$ | $\mathbf{0.1175 \pm 0.0837}$ | **87.50%** |
| 0.1 | HRegNet* | $0.0406 \pm 0.0201$ | $0.1606 \pm 0.1195$ | 95.60% |
| | Ours | $\mathbf{0.0294 \pm 0.0172}$ | $\mathbf{0.1184 \pm 0.0889}$ | **97.93%** |
| 0.2 | HRegNet* | $0.0444 \pm 0.0271$ | $0.1613 \pm 0.1192$ | 97.62% |
| | Ours | $\mathbf{0.0310 \pm 0.0225}$ | $\mathbf{0.1198 \pm 0.0881}$ | **98.13%** |
| 0.3 | HRegNet* | $0.0455 \pm 0.0308$ | $0.1616 \pm 0.1194$ | 99.52% |
| | Ours | $\mathbf{0.0318 \pm 0.0259}$ | $\mathbf{0.1197 \pm 0.0879}$ | **99.88%** |

Table 12: **Comparison results with HRegNet*** (**Lu et al., 2023b**) **under different rotation thresholds** $\theta_R$ **on the KITTI dataset.** $\theta_T$ is set to 0.05m.

| $\theta_R$ (deg) | Method | RTE (m) | RRE (deg) | Recall |
|---|---|---|---|---|
| 0.25 | HRegNet* | $0.0302 \pm 0.0109$ | $0.1210 \pm 0.0627$ | 57.20% |
| | Ours | $\mathbf{0.0241 \pm 0.0109}$ | $\mathbf{0.1006 \pm 0.0538}$ | **81.10%** |
| 0.5 | HRegNet* | $0.0304 \pm 0.0109$ | $0.1524 \pm 0.0973$ | 67.38% |
| | Ours | $\mathbf{0.0242 \pm 0.0109}$ | $\mathbf{0.1158 \pm 0.0781}$ | **87.12%** |
| 1 | HRegNet* | $0.0304 \pm 0.0109$ | $0.1593 \pm 0.1121$ | 68.41% |
| | Ours | $\mathbf{0.0243 \pm 0.0109}$ | $\mathbf{0.1192 \pm 0.0882}$ | **87.71%** |
| 2 | HRegNet* | $0.0304 \pm 0.0109$ | $0.599 \pm 0.1147$ | 68.45% |
| | Ours | $\mathbf{0.0243 \pm 0.0109}$ | $\mathbf{0.1193 \pm 0.0890}$ | **87.72%** |

Table 13: **Ablation studies of temporal encoding methods on KITTI.**

| Model | RTE (m) | RRE (deg) | Recall | Time (ms) |
|---|---|---|---|---|
| GRU | $0.034 \pm 0.039$ | $0.133 \pm 0.115$ | **100%** | 135 |
| Attention | $0.033 \pm 0.039$ | $0.138 \pm 0.121$ | **100%** | 136 |
| Mamba (Ours) | $\mathbf{0.032 \pm 0.036}$ | $\mathbf{0.122 \pm 0.096}$ | **100%** | **133** |

Table 14: **Ablation studies for different Mamba layers on KITTI.**

| Mamba layers | RTE (m) | RRE (deg) | Recall |
|---|---|---|---|
| 1 | **0.032 ± 0.036** | 0.122 ± 0.096 | **100%** |
| 2 | **0.032** ± 0.037 | **0.120** ± 0.103 | **100%** |
| 3 | **0.032** ± 0.037 | 0.121 ± **0.092** | **100%** |

Table 15: **Ablation studies of the learnable parameters ($W_1$, $W_2$) in Dynamic History Weighting.**

| Method | RTE (m) | RRE (deg) | Recall |
|---|---|---|---|
| MORE | **0.032 ± 0.036** | **0.122 ± 0.096** | **100%** |
| MORE w/ fixed $W_1$ | 0.036 ± 0.039 | 0.131 ± 0.103 | **100%** |
| MORE w/ fixed $W_2$ | 0.037 ± 0.041 | 0.134 ± 0.107 | **100%** |

1&2), reducing registration accuracy. Sequential observations help identify the occlusion, as the car reappears in Frame 3 and can be correctly matched with Frame 1.

**Registration Errors.** We display the registration errors of two scenes in Fig. 12. Compared to HRegNet* (Lu et al., 2023b), our method MORE generates much smaller errors thanks to the sufficient exploration of the temporal information.

**Dynamic Weighting Visualization.** We visualize the weighting matrix using PCA and its corresponding 2D trajectories in Fig. 13. From the figure, it is clear that similar historical trajectories often share similar weights, e.g., (a, b, c), (d, e), (f, g, h), and (i, j). This demonstrates the effectiveness of our Dynamic History Weighting method in learning weights from historical poses to influence the current motion features.

**Registration Samples.** We display more visualization about the point cloud registration performance on three large-scale outdoor datasets: KITTI in Fig. 14, nuScenes in Fig. 15, and Apollo-Southbay in Fig. 16. From these figures, it is clear that our proposed method accurately registers input point pairs even though there are large displacements (low-overlap problem).

# F  LLM USAGE STATEMENT

A large language model (ChatGPT) was used only for limited editorial assistance: (i) spelling/grammar checks; (ii) minor wording refinements that did not alter technical content; and (iii) occasional condensation and formatting suggestions. The LLM contributed nothing to the research itself (ideation, methods, experiments, analyses, drafting, or conclusions); it is not an author and assumes no responsibility for the content.

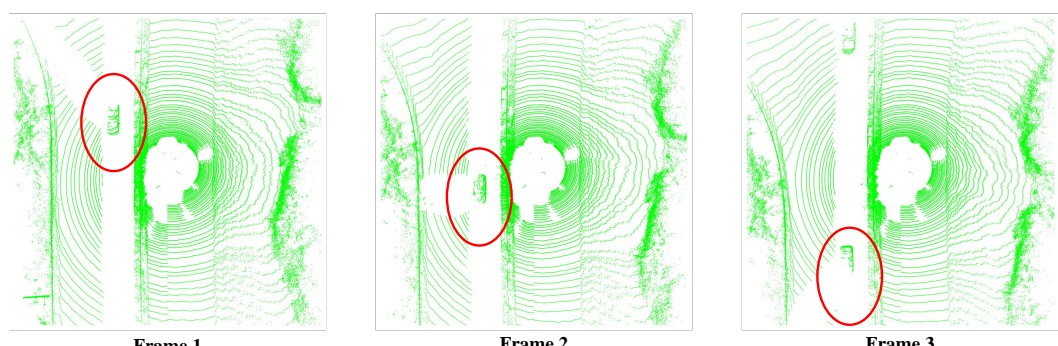

Figure 10: **A dynamic case in KITTI.** In a highway scene, partial observations hinder the recognition of the highly dynamic car.

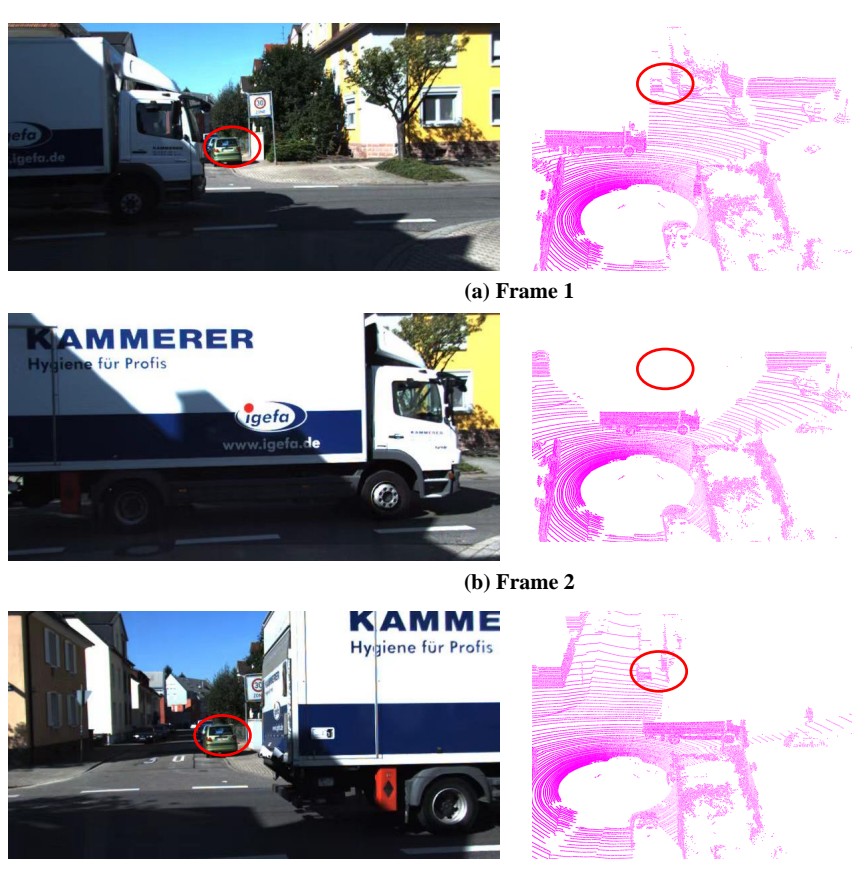

Figure 11: **An occlusion case in KITTI.** A pairwise pose regression can fail in occluded scenarios, e.g., the static car in the red circle in Frame 2, breaking the correlation in Frames 1&2 and 2&3.

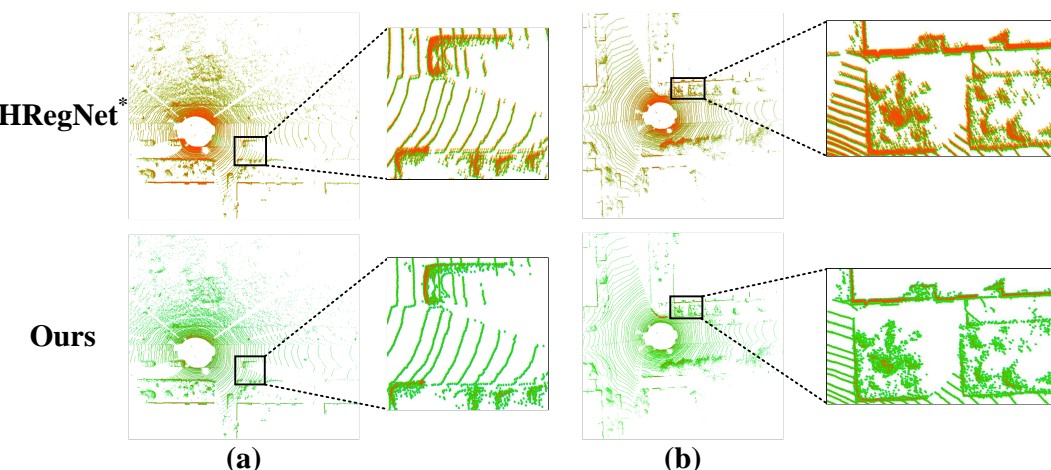

**(a)**          **(b)**

Figure 12: **Registration error visualization.** Our estimation (blue points) shows much smaller errors (red line) compared to HRegNet* (Lu et al., 2023b) (yellow points) relative to GT (green points).

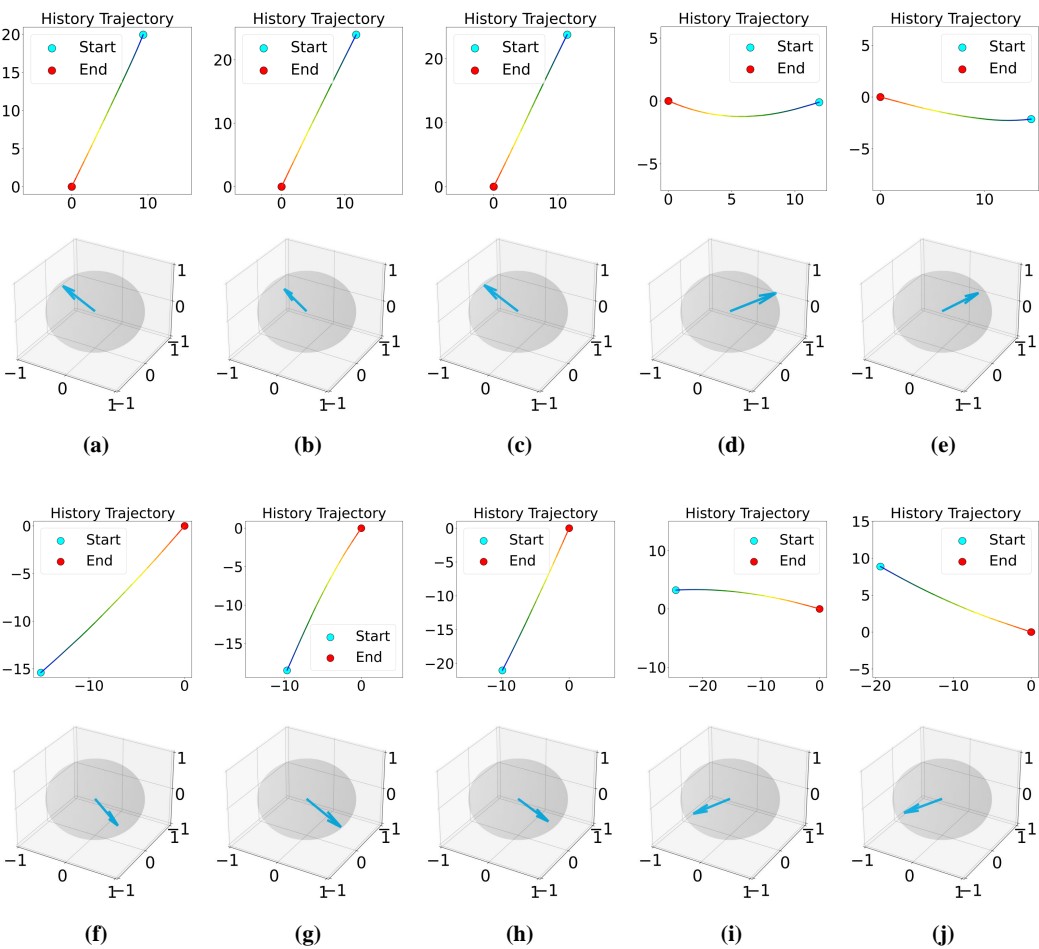

Figure 13: **The visualization of Dynamic History Weighting.** We show some samples about 2D trajectories (upper figures) and their corresponding weights (lower figures), which are generated from the PCA analysis of the weight matrix $W_2$.

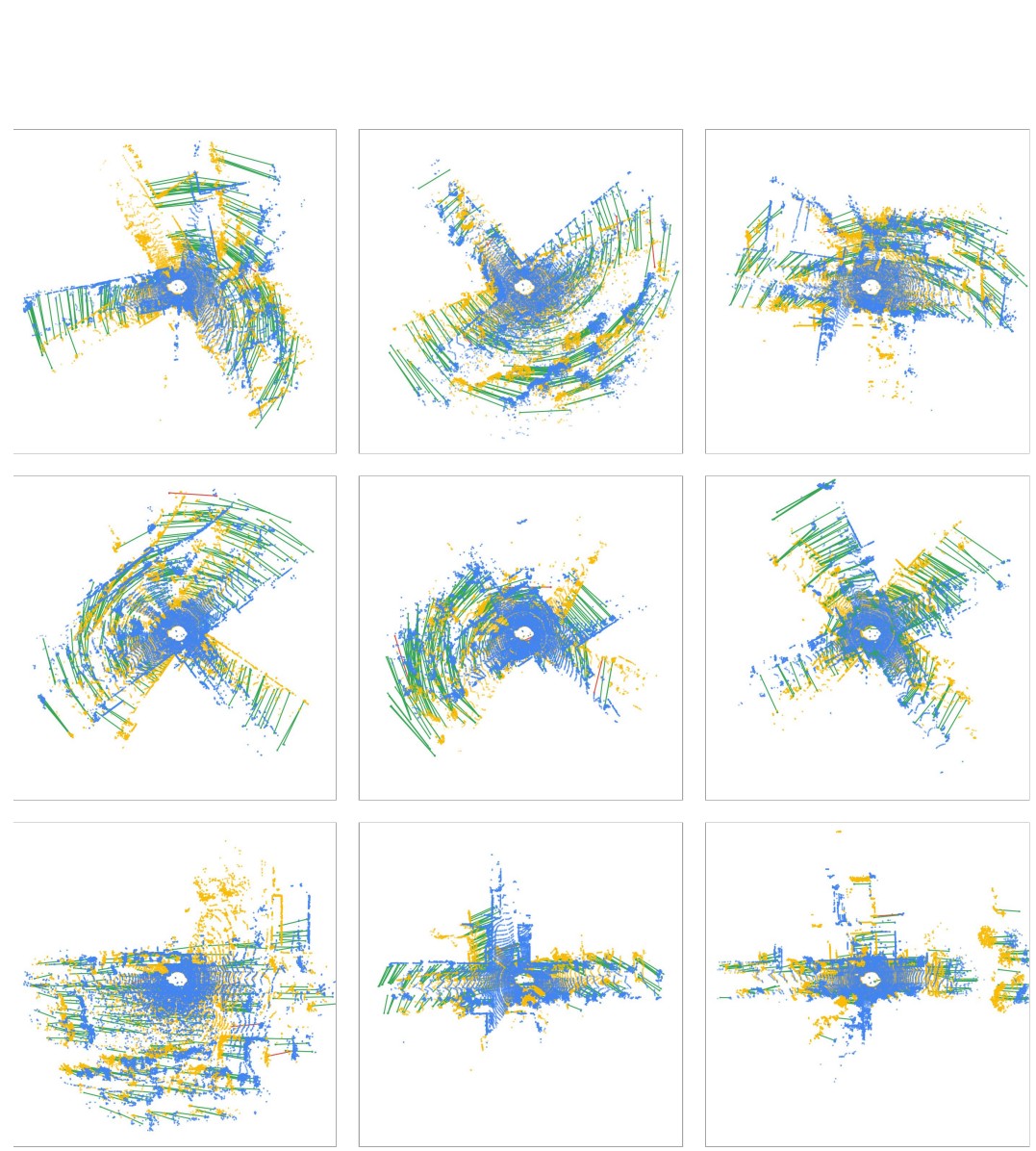

Figure 14: **The visualization of point cloud registration on KITTI.** Yellow and blue points indicate input point pair frames, with green lines showing accurate correspondences (inliers) and red lines showing inaccurate ones (outliers). This figure is best viewed in zoomed-in color mode.

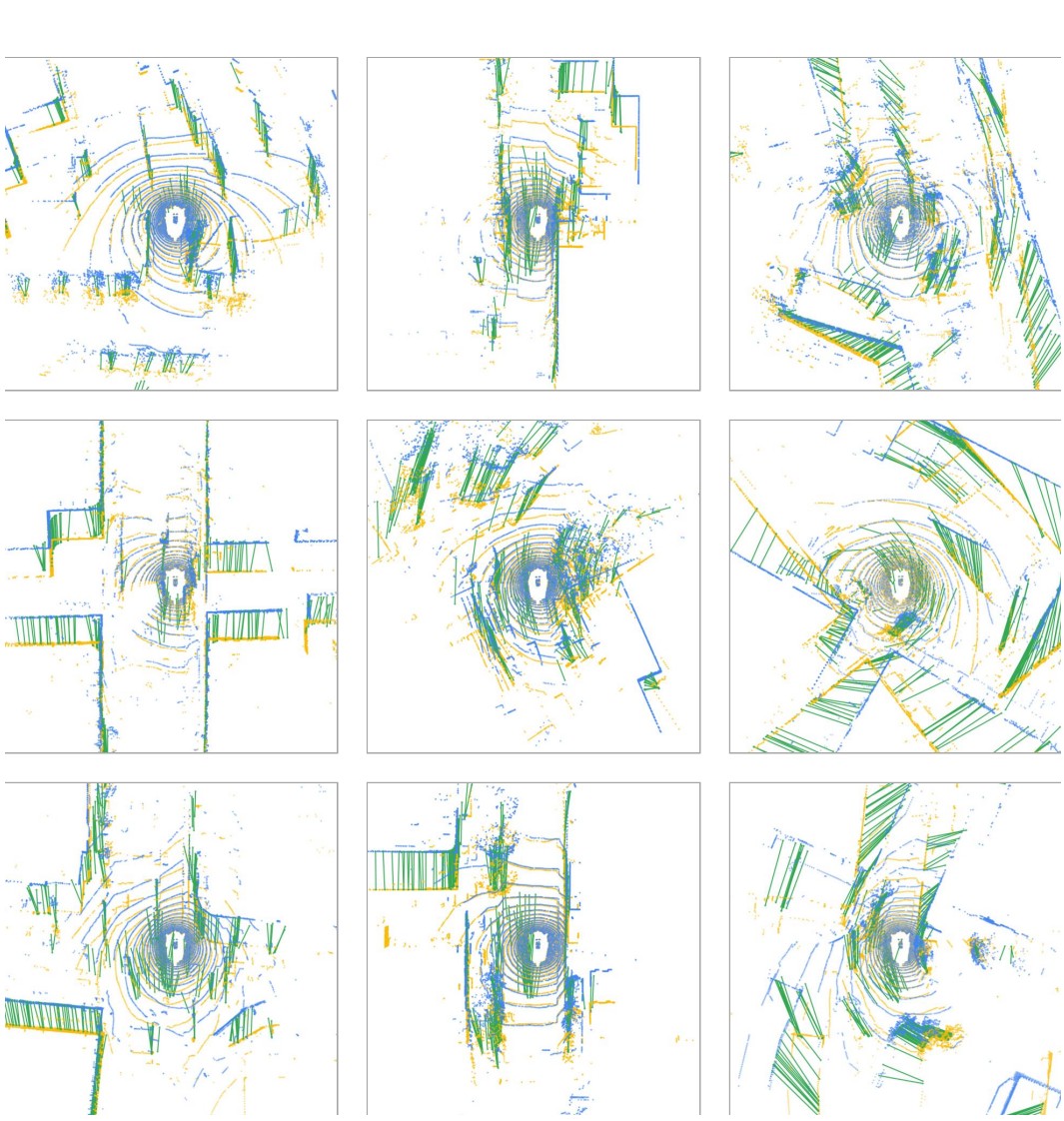

Figure 15: **The visualization of point cloud registration on nuScenes.** Yellow and blue points indicate input point pair frames, with green lines showing accurate correspondences (inliers) and red lines showing inaccurate ones (outliers). This figure is best viewed in zoomed-in color mode.

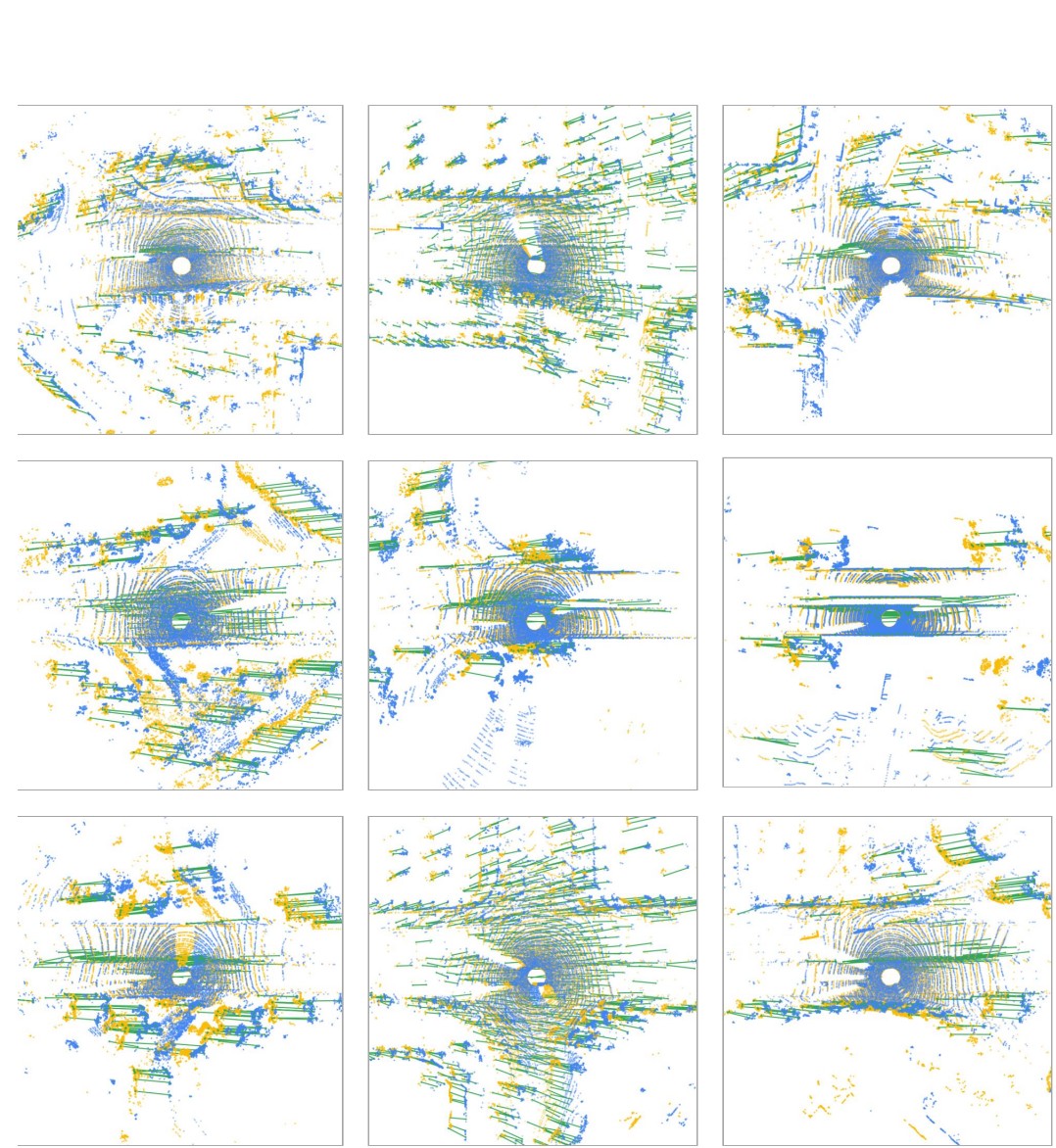

Figure 16: **The visualization of point cloud registration on Apollo-Southbay.** Yellow and blue points indicate input point pair frames, with green lines showing accurate correspondences (inliers) and red lines showing inaccurate ones (outliers). This figure is best viewed in zoomed-in color mode.

