# OpenReview forum: "MORE: Multi-Frame Outdoor Point Cloud Registration with Temporal Memory Buffer"
_ICLR.cc/2026/Conference — ICLR 2026 Conference Withdrawn Submission_

### Official Review · Reviewer_cV2k · 2025-10-26

**Soundness:** 3
**Presentation:** 2
**Contribution:** 2
**Rating:** 4
**Confidence:** 4

**Summary:**

This paper proposes MORE, a framework designed to overcome the limitations of pairwise point cloud registration in outdoor environments. MORE exploits long-term temporal information from consecutive LiDAR sequences to enhance registration robustness and accuracy. The method introduces two complementary memory buffers—an implicit memory feature buffer and an explicit memory pose buffer—to store and dynamically update temporal pose-related features. A Mamba-based temporal encoding layer is employed to integrate current and historical motion features, while a dynamic history weighting module adaptively balances their contributions.

**Strengths:**

1. Leveraging temporal sequence information to improve the accuracy of point cloud registration is reasonable.

2. Evaluated on multiple large-scale datasets and enough ablations, including both outdoor and indoor benchmarks.

3. The paper is clearly written and well-structured.

**Weaknesses:**

1. The use of historical temporal information may introduce error accumulation, since the current transformation correction depends on previous estimations. How the proposed framework mitigates potential drift or compounding errors.

2. All historical information, including feature refinement, appears to be learned entirely through network regression, without an explicit mechanism to model or constrain the interaction between the current state and the historical states. This may limit interpretability and stability.

3. In Equations (9–10), the role and influence of the weighting term w are not clearly explained. Moreover, Figures 5 and 13 are difficult to interpret and do not clearly illustrate the effect of this module.

4. It is also unclear how temporal order is defined in the 3DMatch dataset from unordered or undirected pruned graphs.

5. Under trajectories with strong motion dynamics or abrupt changes, would the use of historical information introduce instability or degrade registration accuracy?

**Questions:**

Please refer to the weaknesses listed above. I will raise my score once my concerns are well-addressed.

---

### Official Review · Reviewer_smTT · 2025-10-26

**Soundness:** 2
**Presentation:** 2
**Contribution:** 2
**Rating:** 2
**Confidence:** 4

**Summary:**

The paper proposes MORE, a LiDAR-based registration framework that augments pairwise registration with temporal memory buffers for feature and pose information. Each new frame pair is firstly registered using a coarse-matching layer as per HRegNet and then refined with dynamic history weighting using both current and historical context over the past T frames. The method aims to improve robustness and accuracy for outdoor LiDAR registration, reporting consistent gains on KITTI, nuScenes, and Apollo-Southbay datasets compared to other pairwise registration baselines and their multi-view versions, e.g., HRegNet$^*$. Generalization capability is demonstrated on the indoor registration task.

**Strengths:**

1. The idea of incorporating temporal priors via memory buffers is conceptually appealing and relevant for sequential LiDAR streams.
2. Quantitative results show steady improvements and generalization over strong baselines. Ablations on various components and hyperparameters help clarify their contributions and validate design choices.

**Weaknesses:**

1. The paper does not explicitly describe what inputs are required at inference. From the method section, the framework seems to require a sequential input to build its temporal buffers, outputting a relative pose for each step. This setup suggests the method is rather suitable for streaming odometry-like scenarios rather than general pairwise registration. It is unclear whether MORE can handle two arbitrary scans without prior temporal context, or how it performs when initialized with empty memory buffers.

2. Because the model relies on temporally adjacent frames, it is conceptually closer to LiDAR odometry than to general registration, yet no comparisons are provided against odometry or SLAM systems, which are believed to be the most relevant baselines for a streaming registration framework. In particular, the concept of a sliding window has been previously used in LiDAR odometry systems and shown to be effective for accurate pose estimation (e.g., BALM [1], LIO-SAM [2]).

3. The proposed pipeline inherits the hierarchical feature encoder and coarse matching from HRegNet. The improvement may largely stem from temporal averaging of HRegNet’s features rather than a fundamentally new registration strategy. An ablation replacing HRegNet with a simpler encoder would better isolate the gain from the temporal-memory module itself.

[1] Liu, Zheng, and Fu Zhang. "Balm: Bundle adjustment for lidar mapping." R-AL 2021
[2] Shan, Tixiao, et al. "Lio-sam: Tightly-coupled lidar inertial odometry via smoothing and mapping." IROS 2020.

**Questions:**

1. In this work, there is no explanation of how rotations are decoded and orthogonalized from the latent feature maps. In Eq. 8, quaternions are concatenated with feature vectors before refinement, but it is unclear why direct concatenation of quaternion parameters with feature embeddings is meaningful.

2. In Tables 1-3, the proposed method is reported to be slightly faster than HRegNet$^*$. Since MORE relies on coarse-matching from HRegNet, integrates additional memory modules, and processes a temporal window of T scans per inference step, such results raise questions. Unless these historical features are fully cached and reused, the additional temporal encoding and memory operations should increase computational cost rather than reduce it. Further clarifications and a per-module timing breakdown are necessary to support the reported gains.

3. The reported point-to-plane ICP performance in Tables 1-3 is surprisingly high, although ICP is known to fail without a good initialization. No specification on how ICP P2Pl was initialized is provided. If ICP used ground-truth initialization, the comparison would not reflect the real difficulty of uninitialized registration.

4. The "scale gap" introduced in line 200 to motivate the dynamic history weighting module is unclear and will benefit from an illustrative example.

5. Can the model handle inverse motion, when the two input point clouds are swapped?

---

### Official Review · Reviewer_n9DG · 2025-11-01

**Soundness:** 3
**Presentation:** 2
**Contribution:** 2
**Rating:** 2
**Confidence:** 4

**Summary:**

This paper presents a multi-frame outdoor point cloud registration network (MORE) that incorporates two temporal memory buffers: an implicit memory feature buffer and an explicit memory pose buffer. The method is evaluated on outdoor and indoor datasets and claims to achieve superior performance.

**Strengths:**

1 The idea of using temporal memory buffers to capture sequential information in point cloud registration is well-motivated and relevant.

2 The method achieves state-of-the-art (SOTA) results on the reported benchmarks, demonstrating its effectiveness.

**Weaknesses:**

1 The method does not explicitly address potential drift accumulation over long sequences. It is unclear whether the memory buffers can effectively mitigate error propagation as the number of frames increases. Ablation studies analyzing performance w.r.t. sequence length would strengthen the validation.

2 The use of a memory-based pose buffer appears functionally similar to the pose graph updating mechanism in methods like Spanner3R [1]. The authors should clearly differentiate their contribution, particularly in how the joint design of implicit and explicit buffersoffers a fundamental advantage over prior memory-aided registration approaches.

[1] 3D Reconstruction with Spatial Memory, 3DV 2025

**Questions:**

1 When comparing against other methods, such as on KITTI, were the baselines also trained on sequences 00-05, or was a different data split used? If baselines were trained on different splits or even different datasets, the comparison may be unfair, and the authors should clarify this to ensure a valid evaluation.

2 To better assess generalization, could the authors perform cross-dataset experiments? For example, training on KITTI and testing on nuScenes and Apollo-Southbay would more rigorously demonstrate the robustness of the proposed memory buffers.

---

### Official Review · Reviewer_UR6B · 2025-11-09

**Soundness:** 3
**Presentation:** 3
**Contribution:** 2
**Rating:** 4
**Confidence:** 4

**Summary:**

This paper proposes MORE, a novel framework for multi-frame outdoor point cloud registration. The authors argue that existing pairwise registration methods neglect the rich temporal information inherent in LiDAR sequences. To address this, MORE leverages temporal memory to enhance the current pairwise registration task. The core of the method involves two memory buffers: a "Memory Feature Buffer" (MFB) to store implicit motion-related features, and a "Memory Pose Buffer" (MPB) to store explicit historical poses. The framework employs a Mamba-based encoder for efficient temporal fusion and a "Dynamic History Weighting" (DHW) module to adaptively merge features from these different sources. Experiments show that the method achieves state-of-the-art (SOTA) performance on three large-scale outdoor datasets: KITTI, nuScenes, and Apollo-Southbay.

**Strengths:**

1. The paper introduces a novel framework for multi-frame point cloud registration. A key strength is its explicit modeling of temporal features from sequential point clouds. The proposed memory buffer mechanism (MFB and MPB) provides a principled way to store, update, and leverage this temporal information, which is then used to effectively enhance the robustness and accuracy of the underlying pairwise registration task.

2. The method achieves state-of-the-art (SOTA) performance on several large-scale benchmark datasets (e.g., KITTI, nuScenes). This empirical success is well-supported by a comprehensive set of supplementary experiments and detailed ablation studies, which rigorously validate the effectiveness of the individual components of the proposed method.

**Weaknesses:**

1. The proposed framework is, in essence, a temporal module built on top of an existing pairwise registration method. Given that the multi-frame input provides access to significantly more information, a performance improvement over pairwise-only baselines is intuitive and largely expected. This makes the direct comparison to pairwise-only methods seem unfair and potentially overstates the novel contribution. To truly isolate and demonstrate the superiority of the proposed temporal module, the authors should include comparisons against other multi-frame baselines, such as state-of-the-art LiDAR-SLAM/Odometry front-ends (which also leverage temporal data) or multi-frame optimization-based registration methods.

2. The design choices for the multi-frame feature fusion and the temporal module lack sufficient justification and interpretability. For instance, feature fusion merely concatenates features and poses from different frames without alignment, and the paper in the main text lacks a clear explanation for selecting Mamba.

3. The performance on the chosen datasets appears to be saturated, with the method achieving a 100% recall on all benchmarks under standard thresholds. The advantage of the curve in Figure 3 is also very slight. The authors should evaluate on more challenging datasets or perform a deep analysis by mining hard samples. Additionally, the paper is missing comparisons against several more recent and highly relevant baselines, such as GeoTransformer, MAC, and PARE-Net, to better contextualize its performance.

**Questions:**

1. As mentioned in the weaknesses section, providing more baseline comparisons (including updated work and multi-frame baselines) along with more compelling comparisons on challenging datasets would more effectively demonstrate the effectiveness of the proposed method.

2. Regarding the MFB method, this approach concatenates “cross-frame features” $F_t$ from different timestamps. Are these features $F_t$ transform-invariant? Is it methodologically sound to concatenate unaligned features directly? Although the ablation analysis of the mamba module is listed in the appendix, could authors elaborate further on the motivation for using mamba in the main text?

3. In Table 5, the baseline with both MFB and MPB removed appears to have worse performance than the HRegNet* baseline reported in Table 1. What are the precise architectural differences between the Table 5 baseline (MFB/MPB removed) and the HRegNet* baseline from Table 1?

4. For the ablation in Table 7, what is the exact implementation of the "w/o DHW" baseline? How are the temporal features from MFB and MPB fused without DHW module? Why did this baseline significantly outperform previous methods?

5. Why does the method in this paper achieve faster inference times on Table1 and Table2 compared to HRegNet* when using multi-frame sequences and memory?

6. Since multi-frame sequences are required, how is the buffer effectively managed during the first frame (or first few frames) registration? Does directly filling the buffer have any impact on actual performance? Any quantitative comparison experiments?


Typos
L370：Fig.16 -> Fig.5
L422：Table14 -> Table7

---

### Note · Authors · 2025-11-12

I have read and agree with the venue's withdrawal policy on behalf of myself and my co-authors.